# WHEN BIGGER ISN'T BETTER: THE ROLE OF MODEL-DATA COMPLEXITY IN TIME SERIES FORECASTS

## ABSTRACT

Large, over-parameterized models have become the dominant paradigm in machine learning, with foundation models claiming universal applicability across diverse tasks such as time series forecasting. Yet, it remains unclear how such models behave across the full spectrum of data complexity- as estimated by the complexity of the generative processes that produce them. In this work, we show that large foundation models often struggle with forecasting on simple data. We propose a systematic benchmarking approach to evaluate models at multiple levels of complexity, from classic statistical to foundation models, against datasets spanning from simple, deterministic patterns to highly stochastic processes. By evaluating models that range from classic statistical methods (e.g., ARIMA) through mid-complexity deep networks to large foundational models, we show that model effectiveness depends jointly on model complexity and data complexity. Simpler, structured datasets often favor lower-capacity or classical methods, while complex, noisy datasets generally benefit from higher-capacity machine learning models. These results highlight the importance of task-specific model selection, balancing data and model complexity. In contrast, foundation models often fail on simple signals where inductive bias and parsimonious modeling are sufficient. These findings show that "bigger" is not inherently "better," reaffirming the classical approximation–estimation trade-offs in the zero-shot setting, and underscore the need for data-aware model selection rather than one-size-fits-all deployment.

## 1 INTRODUCTION

Time series forecasting with foundation models, inspired by large language models (LLMS) and the transformer architecture, has emerged as a leading paradigm in modern machine learning (Ansari et al., 2024; Woo et al., 2024; Das et al., 2024; Rasul et al., 2024; Jin et al., 2024). These models, comprising millions or billions of parameters and trained on massive heterogeneous datasets, promise zero-shot generalization to unseen time series across healthcare, finance, energy, and climate domains. However, the benchmarking philosophy borrowed from language modeling (where "more diverse data and larger models yield better performance") still has not proven to be fully aligned with the nature of time series data.

Unlike language, where even simple sentences embody complex grammatical structures and semantic relationships that benefit from overparametrization, time series often originate from well-understood dynamical processes with strong inductive biases. For example, a periodic signal like the sine wave, governed by deterministic mathematical rules, is analogue to a standard subject-verb-object sentence. Yet, while LLMs excel at such simple sentences, we find that time series foundation models do not outperform classic statistical modeling on elementary signals. This discrepancy reveals a gap: whereas language models successfully leverage their over-parameterization to handle both simple and complex linguistic structures, time series foundation models do not always learn to recognize basic deterministic patterns that any human expert could immediately identify through basic analysis.

In addition, traditional scientific modeling of complex dynamical systems has long relied on simple functional forms such as sinusoids and polynomials. In fact, domain expertise guides model selection through inductive biases: smoothness assumptions, periodicity, trend-cycle decompositions, and physical constraints (Box et al., 2015). These intuitive choices often outperform black-box

Table 1: Models used in study organized by model complexity

| Statistical Models | Deep Learning Models | | Foundation Models |
|---|---|---|---|
| ARIMA | DeepAR | DeepState | Chronos |
| NARMA | Informer | PatchTST | Moirai |
| | TimesNet | TFT | TimesFM |
| | NBeats | NHits | |
| | NLinear | | |

approaches by incorporating the correct structural assumptions about the data-generating process. Thus, the under-performance of foundation models on simple datasets highlights their limitation in encoding simple patterns. Investigating the models' prediction across a spectrum of data complexity, from simple patterns to highly stochastic data, becomes crucial to gain understanding into their capabilities.

In this work, we systematically evaluate how model complexity interacts with data complexity in time series forecasting, where we define data complexity through the complexity of the underlying generating process. Our results show that while foundation models outperform classical methods like ARIMA on highly stochastic (complex) data, they struggle with forecasting on simple patterns (low complexity) like sine waves and polynomial trends. These findings highlight the relevance of the approximation-estimation trade-off from classical learning theory (Hastie et al., 2009) in the age of over-parameterized models. In such cases, complex models misallocate their representational capacity, fitting to noise that does not exist in simple data, and generalize poorly, contradicting theoretical results like double descent (Belkin et al., 2019).

Our benchmark highlights a limitation in foundation models and highlights the need for improvements in their scientific applications, where recognizing and extrapolating simple patterns is often more valuable than fitting complex, noisy real-world data.Indeed, the simplicity and parsimony of governing equations as the source of data generation has largely driven scientific discovery in the last few centuries (Kutz & Brunton, 2022). We argue that model selection for time series should not default to the largest available model but explicitly consider the complexity of the underlying data-generating process. Our framework provides practitioners with a systematic approach to assess when foundation models offer genuine advantages versus when simple theory-backed methods remain superior.

## 2 BENCHMARKING TIME FORECASTING MODELS

Our benchmarking framework classifies time forecasting models by their model complexity and evaluate their zero-shot capability on data sets with varying data complexity. We define model complexity by the number of parameters in the model; while, data complexity is determined by the complexity of the underlying generating process. We measure the model's performance by the root mean squared error between the predicted forecast and the true time series.

### 2.1 MODEL COMPLEXITY

When using the number of parameters to measure model complexity, we can broadly categorize forecasting models into three broad classes: statistical models, deep learning models, and foundation models (Table 1). Statistical models include classical forecasting methods like ARIMA and NARMAX, which do not require a training phase in the conventional sense. Instead, they fit a relatively small number of parameters to the historical data in order to generate a future forecast. In contrast, deep learning models are based on machine learning architectures like convolution neural nets (Wu et al., 2023), recurrent neural nets (Salinas et al., 2020; Rangapuram et al., 2023), or transformers and attention mechanisms (Zhou et al., 2021; Wu et al., 2021; Lim et al., 2021). Typically comprising of hundred thousands to less than ten million parameters, these models require training and perform best when fine-tuned for a specific task, though most claim some level of zero-shot capability. Finally, foundation models for time forecasting are commonly adapted from LLMs. They

Table 2: Datasets used to train deep learning models

| Domain | Dataset Name | Number of Time Series |
|--------|--------------|----------------------|
| Energy | London Smart Meters | 5560 |
| Transport | Traffic | 862 |
| | Uber TLC Hourly | 262 |
| | Uber TLC Daily | |
| Finance | M4 Hourly | 414 |
| | M4 Daily | 4227 |
| | M4 Monthly | 48000 |
| Web | Wiki Rolling | 9535 |
| Nature | KDD Cup 2018 | 270 |

are trained on massive datasets and designed for zero-shot forecasting (Woo et al., 2024; Ansari et al., 2024; Das et al., 2024). The foundation models selected for this study - Moirai, Chronos, and Timesfm - are comprised of 91 million, 200 million, and 500 million parameters, respectively.

## 2.2 DATA COMPLEXITY

We define data complexity in terms of the complexity of the process that generates the time series. This perspective connects to foundational ideas in information theory and statistical learning, where complexity has been formalized through notions such as Kolmogorov complexity (Kolmogorov, 1965), Minimum Description Length (MDL; Rissanen, 1978; 1998), and measures from dynamical systems such as statistical complexity (Crutchfield & Young, 1989), Lempel–Ziv complexity (Lempel & Ziv, 2003), and permutation entropy (Bandt & Pompe, 2002). These frameworks emphasize that a dataset is "simple" if it can be generated or compressed by a concise rule or model, and "complex" if it requires a longer description or encodes high-dimensional hidden dependencies.

Following this principle, we categorize datasets by the known or presumed complexity of their generators. Synthetic datasets generated from simple, deterministic functions are considered *low complexity data*. For example, sine waves, elliptic functions, and Chebyshev polynomials can be described by a small number of parameters and exhibit regular, noise-free structure. Such signals are well matched to statistical methods with strong inductive biases (e.g., ARIMA, Fourier methods). Real-world datasets that retain some deterministic structure but include noise and variability are considered *medium complexity data*. For instance, electricity consumption data (Jensen et al., 2017) is governed by predictable cycles (daily, weekly) but influenced by stochastic fluctuations (weather, human activity). This type of data typically benefits from expert inductive biases, but might contain hidden variables that might defy simple statistical fits. Stochastic datasets with hidden variables and strong noise components, approximating high-dimensional systems and projected on one dimension, are considered *high complexity data*. Stock market indices such as the S&P 500 are canonical examples, where dynamics reflect the interaction of countless hidden agents and exogenous inputs and forcings.

This framing allows us to construct a benchmark where models of varying complexity are tested against datasets whose complexity is controlled, or at least qualitatively understood. Unlike most existing benchmarks, which emphasize performance on heterogeneous real-world data, our approach directly probes the relationship between model complexity and data complexity. In doing so, we highlight when increased model capacity is advantageous and when it fails on deceptively simple signals.

## 3 METHODS

### 3.1 TRAINING DEEP LEARNING MODELS

To enable comparison on zero-shot forecasting, it is necessary to train the deep learning models on a dataset similar to the training corpora used for foundation models. We compiled training datasets

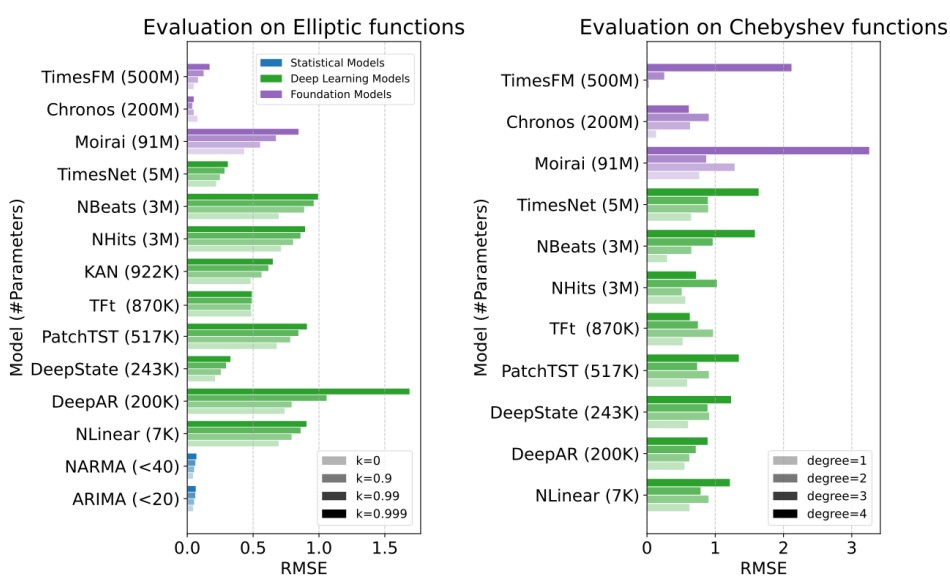

Figure 1: RMSE (root mean squared error) of selected time forecasting models for low complexity data: elliptic functions (left) and Chebyshev polynomials (right).

across five domains: energy, transportation, finance, web, and nature, which overlap with those used to train the selected foundation models. The specific datasets for each domain are detailed in table 2. While some of these datasets are explicitly included in the training corpora for a few foundation models, some are only domain-aligned but not identical. All datasets are publicly available through the GluonTS repository (Alexandrov et al., 2020). Deep learning models were retrieved from Nixtla and trained in all instances of the training data set (Table 2).

### 3.2 Hyperparameters for Statistical Models

While the statistical models like ARIMA and NARMAX are fitted to a given context, the choice of hyperparameters, such as the number of lags, order of difference, and degree of polynomial basis, can improve the model performance. To this end, for each dataset to zero-shot forecast, we subjected ARIMA and NARMA to a grid search for the best set of hyperparameters using a held out time series from the same dataset. The collection of hyperparameters that achieved the lowest RMSE was chosen to perform zero shot forecasting on the unseen test set.

## 4 Results

Our statistical models typically performed better for low complexity data (figure 1). The statistical models performed strongly on elliptic functions, which are a generalization of purely sinusoidal behavior, and achieved almost zero RMSE on Chebyshev polynomials (see table 3). Among the foundation models, Chronos and TimesFM performed almost as well as the statistical models on the elliptic functions, but lagged behind ARIMA and NARMA on the Chebyshev polynomials. Meanwhile, some deep learning models, such as DeepState and TimesNet, were able to outperform Moirai on elliptic data but also struggled on the Chebyshev polynomials. The deep learning models generally did as well as the foundation models on the Chebyshev polynomials but achieved better results at the higher degrees. Notably, forecasting became more difficult for Moirai and TimesFM with higher elliptic constants $k$, and similarly, their performance degraded as the degree of Chebyshev polynomials increased. We show some of these forecasts explicitly in figures 2 and 3.

In contrast, the foundation models generally outperformed the statistical models on real-world data (figure 4). Chronos and TimesFM achieved the lowest RMSE on the electric grid data even though Moirai was still outperformed by several deep learning models, such as temporal fusion transformer

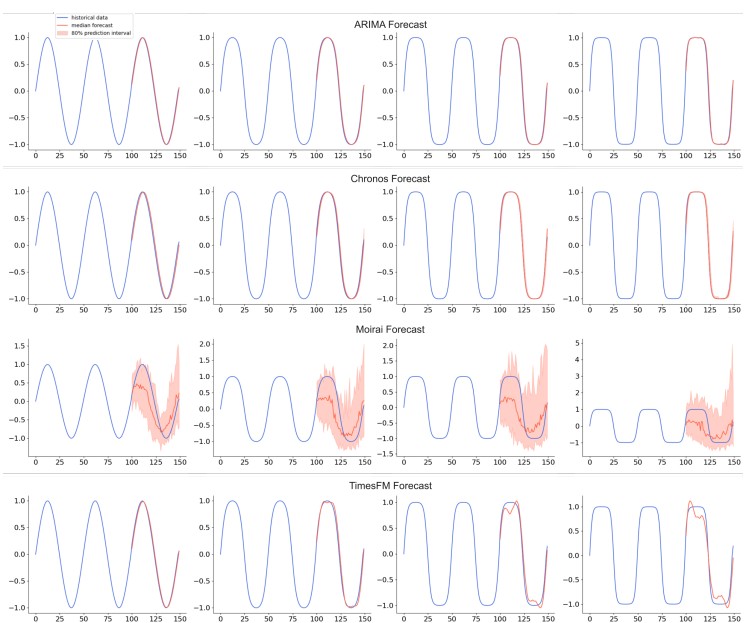

Figure 2: Jacobian elliptic forecasts from ARIMA, Chronos, Moirai, and TimesFM (top to bottom row respectively). The 80% confidence interval is shown for Chronos and Moirai. From left to right, the free parameter on the Jacobian elliptic function, $k$, increases in value between 0 and 1.

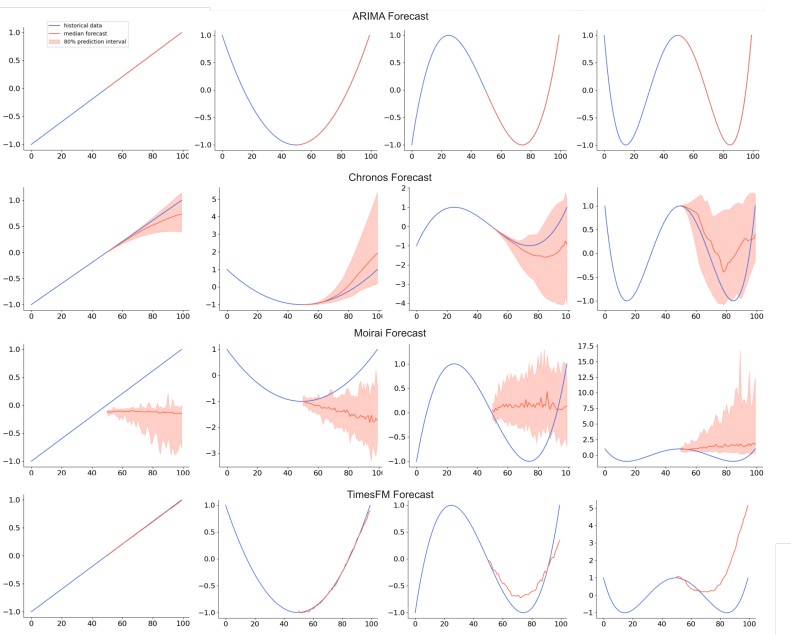

Figure 3: Chebyshev polynomial forecasts from ARIMA, Chronos, Moirai, and TimesFM (top to bottom row respectively). The 80% confidence interval is shown for Chronos and Moirai. From left to right, the degree of the Chebyshev polynomial is increased from 1 to 4.

(TFT), TimesNet, and DeepState. Suprisingly, ARIMA and NARMA remained competitive on this dataset, outperforming the deep learning models. On the highly complex S&P 500 dataset, the statistical models performed poorly and ranked near the bottom (figure 4). Notably, the three foundation models and most of the deep learning models performed similarly, but figure 5 shows

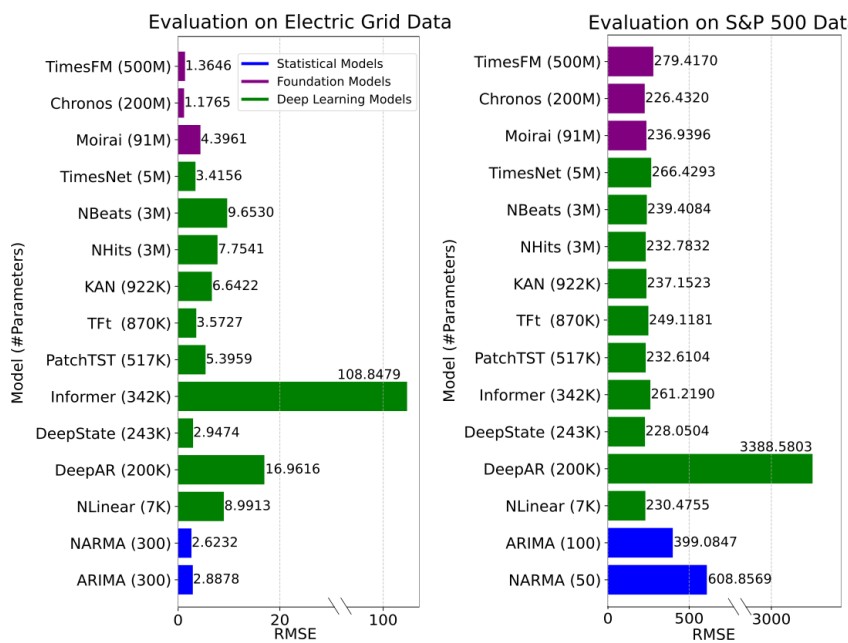

Figure 4: RMSE (root mean squared error) of time forecasting models for (left) moderate complexity data, hourly electric grid consumption over 21 days, and (right) high complexity data, closing prices of S&P 500 over 300 days in 2022.

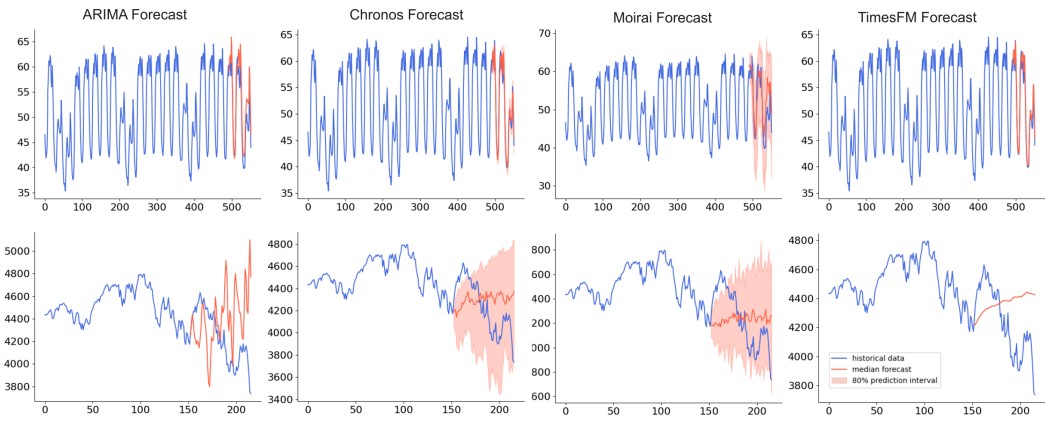

Figure 5: (from left to right) ARIMA, Chronos, Moirai, and TimesFM forecasting on the hourly electric grid consumption over 21 days (top row) and the closing prices of the S&P 500 over 300 days in 2022. (bottom row)

that the probabilistic forecast from Chronos and Moirai does highlight their potential in capturing noisy data accurately. Two of the deep learning model, Informer and DeepAR, failed dramatically on the electric grid and the S&P 500 datasets respectively, indicating instability in their model and possible overfitting. The complete results in tables 3, 4, and 5 also show that Informer achieved some of the worst performance for most datasets.

## 5 DISCUSSION

In our systematic evaluation of time series forecasting models against datasets of varying data complexity, we see that large foundation models have an advantage on noisy complex data but do not

Table 3: Zero shot RMSE of statistical and foundation models

| Dataset | | Statistical Models | | Foundation Models | | |
| --- | --- | --- | --- | --- | --- | --- |
| | | ARIMA | NARMA | Chronos | Moirai | TimesFM |
| Elliptic | 0 | 0.0440 | 0.0440 | 0.0780 | 0.4300 | 0.0490 |
| | 0.9 | 0.0520 | 0.0520 | 0.0490 | 0.5500 | 0.0810 |
| | 0.99 | 0.0610 | 0.0610 | 0.0380 | 0.6700 | 0.1200 |
| | 0.999 | 0.0640 | 0.0700 | 0.0490 | 0.8500 | 0.1700 |
| Chebyshev | 1 | 4.3e-16 | 4.3e-16 | 0.1300 | 0.7700 | 0.0065 |
| | 2 | 3.4e-14 | 3.3e-14 | 0.6300 | 1.3000 | 0.0280 |
| | 3 | 7.1e-13 | 1.0e-12 | 0.9000 | 0.8700 | 0.2500 |
| | 4 | 2.1e-11 | 3.4e-11 | 0.6100 | 3.3000 | 2.1000 |
| Electric Grid | | 2.9000 | 2.6000 | 1.2000 | 4.4000 | 1.4000 |
| S&P 500 | | 400.0000 | 610.0000 | 230.0000 | 240.0000 | 280.0000 |

Table 4: Zero shot RMSE of deep learning models (part 1)

| Dataset | | DeepAR | DeepState | Informer | KAN | NBeats |
| --- | --- | --- | --- | --- | --- | --- |
| Elliptic | 0 | 0.7400 | 0.2100 | 5.4102 | 0.4809 | 0.6948 |
| | 0.9 | 0.7900 | 0.2600 | 5.3360 | 0.5643 | 0.8876 |
| | 0.99 | 1.1000 | 0.2900 | 5.5830 | 0.6158 | 0.9601 |
| | 0.999 | 1.7000 | 0.3300 | 5.6758 | 0.6496 | 0.9931 |
| Chebyshev | 1 | 0.5500 | 0.6000 | 3.6185 | 8.0345 | 0.2947 |
| | 2 | 0.6200 | 0.9100 | 5.7404 | 1.0074 | 0.6502 |
| | 3 | 0.7100 | 0.8900 | 5.7353 | 1.0646 | 0.9630 |
| | 4 | 0.8900 | 1.2000 | 5.1878 | 2.9148 | 1.5811 |
| Electric Grid | | 17.0000 | 2.9000 | 108.8479 | 6.6422 | 9.6530 |
| S&P 500 | | 3400.0000 | 230.0000 | 261.2190 | 237.1523 | 239.4084 |

necessarily outperform simpler models on low complexity data. Their advantage on noisy data can be explained by their extensive training on a large and diverse corpus of real world data. Their under-performance on simple data, however, suggest a limited ability to capture or exploit the underlying structure and smoothness of such signals.

This phenomenon can be explained by the approximation-estimate trade-off. Since foundation models are designed with high representation capacity in order to enable low approximation error across a range of problems, they struggle when the problem given is highly structured, resulting in a higher estimation error. When it comes to deterministic time series, which have strong inductive biases and low variance, simpler models can achieve lower total error by having a smaller capacity. For example, Chronos and TimesFM achieved a smaller RMSE than the statistical models on the electric grid data; yet, they, especially TimesFM, did not consistently outperform ARIMA or NARMA on the elliptic data, despite both exhibiting cyclical patterns. The key difference lies in the noise level, where the electric grid data is noisy and irregular, while the elliptic functions are smooth and deterministic. This disparity in performance suggests their over-parametrization architecture obscures their ability to recognize structural patterns.

Our results also recognize that model performance differed from dataset to dataset. Informer was terribly unstable for forecasting on the electric grid data but performed average for S&P 500. TimesFM, Chronos, TimesNet, and DeepState all seem to forecast well for cyclical data (elliptic functions and electric grid) but do not achieve the best results for noncyclic data.

Some of the limitations of foundation models can be explained by their architectural biases. For example, ARIMA and NARMA easily achieved near-zero error on the Chebyshev polynomials. Meanwhile, Chronos and Moirai both struggle while TimesFM can sometimes some accurate forecasts (at lower degrees of the polynomial). Chronos is encoded to make prediction within the range

Table 5: Zero shot RMSE of deep learning models (part 2)

| Dataset | | NHits | NLinear | PatchTST | TFT | TimesNet |
|---|---|---|---|---|---|---|
| Elliptic | 0 | 0.7124 | 0.6943 | 0.6794 | 0.4861 | 0.2189 |
| | 0.9 | 0.8038 | 0.7923 | 0.7815 | 0.4817 | 0.2475 |
| | 0.99 | 0.8596 | 0.8611 | 0.8441 | 0.4898 | 0.2831 |
| | 0.999 | 0.8943 | 0.9053 | 0.9087 | 0.4901 | 0.3084 |
| Chebyshev | 1 | 0.5619 | 0.6230 | 0.5896 | 0.5224 | 0.6456 |
| | 2 | 0.5086 | 0.9016 | 0.9067 | 0.9684 | 0.8967 |
| | 3 | 1.0249 | 0.7858 | 0.7350 | 0.7459 | 0.8925 |
| | 4 | 0.7192 | 1.2148 | 1.3459 | 0.6274 | 1.6365 |
| Electric Grid | | 7.7541 | 8.9913 | 5.3959 | 3.5727 | 3.4156 |
| S&P 500 | | 232.7832 | 230.4755 | 232.6104 | 249.1181 | 266.4293 |

of the context it is given, which would explain its struggles with extrapolation in trends. Similarly, Moirai tends to make noisy forecasts since it is designed to output from mixed distributions.

## 6  CONCLUSION

We examine the performance of various models, ranging from simple statistical models to highly parametrized foundation models, on datasets with varying complexities. We find that in general, while foundation models have an advantage on noisy data, they struggle with simple, deterministic (low complexity) data. Meanwhile, some deep learning models, trained on considerably less data than the foundation models can perform just as well or better than some foundation models. Our work does show that foundation models have their place in having the best performance for a specific level of data complexity, specifically for high complexity data.

The probabilistic forecasting from Chronos and Moirai makes them both desirable for highly stochastic data. Meanwhile, TimesFM can forecast polynomials to some extent. This benchmark framework should guide practitioners on model selection for real world deployment.

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

# A APPENDIX

## A.1 LARGE LANGUAGE MODEL USAGE

Large language models were used in this work to improve grammar and polish writing. They also assisted in generating codes for experiments and plotting.

