# OpenReview forum: "When Bigger isn't Better: The Role of Model-Data Complexity in Time Series Forecasts"
_ICLR.cc/2026/Conference — ICLR 2026 Conference Withdrawn Submission_

### Official Review · Reviewer_Tr8z · 2025-10-15

**Soundness:** 2
**Presentation:** 3
**Contribution:** 2
**Rating:** 4
**Confidence:** 3

**Summary:**

This paper compares traditional staistical methods (e.g. NARMA, ARIMA, etc.), deep learning models (e.g. PatchTST) and Foundational Time Series Models (e.g. TimesFM, Chronos) on severl time series prediction tasks. These tasks include, (1) simple time series like Jacobian elliptic and Chebyshev polynomials; (2) more complicated tasks like Eletric Grid and S&P 500. The authors find that for simple tasks, statistical methods work best, followed by DL methods and foundational models work worst; for complicated tasks tested, DL methods and foundational models seem to work well. Thus, authors summarize their findings as statistical methods work well for simple and regular time series, while foundation models work for more complicated tasks.

**Strengths:**

The authos directly compare statistical methods to DL-based methods and time series foundational methods, revealing that some newly-proposed DL methods and time series foundational methods are not good. This points out some weakness of foundational models

**Weaknesses:**

1. The observation that foundational models work well on S&P might just be a result of data contamination. (1) Is S&P 500 in training set of foundation models? (2) Even if S&P is not in training set, is other related time series, e.g. DAX, in the training set of such foundation models? The correlation of these index would lead to data leakage issue.
2. The evaluated datasets seem a bit limited. Only several time series are used for evaluating.
3. There have been existing discussions of problems for Time series foundation models. For example, in NeurIPS 2024 time series in the age of large models workshop invited talk by Christoph Bergmeir - Fundamental limitations of foundational forecasting models: The need for multimodality and rigorous evaluation (https://neurips.cc/virtual/2024/108471), the speaker says that time series alone lacks contextual information.

**Questions:**

Since this paper seems a bit similar to position papers, I'd like to propose some questions on behalf a position paper reviewer.

That is, my position agains time series foundation models is like: I think time series along lack too much information for any foundation model to predict its future. For example, if I give you a time series curve from S&P, one curve from some electric factory, one curve from perhaps the infected number of some disease: they might look similar in a period of time. It doesn't make sense to use a so-called foundation model to predict them. We need more context, for example, more alphas for predicting stock price (constructed from order books or news), detailed electricity properties; and perhaps some medical information and differential equation to solve for number of infected people.

What's your opinion on this argument? I'm looking forward to further discussions.

---

### Official Review · Reviewer_kJrX · 2025-10-30

**Soundness:** 1
**Presentation:** 2
**Contribution:** 1
**Rating:** 0
**Confidence:** 5

**Summary:**

This paper shows that time series foundation models underperform classical methods like ARIMA on simple deterministic data but excel on complex noisy data. The finding that overparameterized models struggle with structured simple data is unsurprising and well-known from learning theory. The work provides useful systematic benchmarking but offers limited novelty—essentially confirming that model selection should match data complexity, which is hardly a new insight.

**Strengths:**

- The benchmarking framework is systematic.
- It tests statistical models (ARIMA, NARMA), deep learning models (10+ architectures), and foundation models (Chronos, Moirai, TimesFM) under consistent zero-shot conditions. This is quite comprehensive.

**Weaknesses:**

- The definition of 'complexity' is vague. It is qualitative and subjective rather than rigorously quantified through formal measures (despite citing Kolmogorov complexity, MDL, etc.).
- The novelty is **very limited**. The core finding—that overparameterized models struggle with simple structured data—directly follows from classical bias-variance trade-offs and is well-established in learning theory. It is also already stated in some existing papers (for example 'Scaling Law for Time Series Forecasting').
- It only tests on two real-world datasets which is limited.
- The synthetic data is too simple. Chebyshev polynomials and elliptic functions may not represent the "simple but important" patterns in actual scientific applications.
- Practitioners must know data complexity a priori to select models, but complexity assessment itself is non-trivial. So it hardly helps practitioners.
- It identifies problems but offers no solutions for improving foundation models' performance on structured data.

**Questions:**

If practitioners need to assess data complexity a priori to select the appropriate model (as your framework suggests), but you provide no quantitative method for measuring complexity beyond qualitative categorization, how is this approach practically actionable? In other words, doesn't your framework require solving the very problem it aims to address—knowing which model is suitable before you can choose it?

---

### Official Review · Reviewer_PX4t · 2025-10-30

**Soundness:** 3
**Presentation:** 3
**Contribution:** 3
**Rating:** 6
**Confidence:** 4

**Summary:**

This work benchmarks different types of time-series forecasting models, including statistical models, neural forecasting models, and pre-trained foundation models, and discloses that although some time-series foundation models excel at some data scenarions, they may encounter severe troubles when dealing with extremely simple time series.

**Strengths:**

- Unveiling the weakness of time-series foundation models
- Highlighting the importance of task-specific model selection

**Weaknesses:**

- The weakness of time-series foundation models is demonstrated mostly in synthetic cases. Is there any real-world case that induce stupid mistakes from general time-series foundation models? The paper's insight is very good, but it may need more experiments to cover more real-world datasets, telling us in which real-world cases, some time-series foundation models should be preferred, and in which cases we should avoid using them.

**Questions:**

See the weakness part.

---

### Official Review · Reviewer_WkeU · 2025-10-31

**Soundness:** 2
**Presentation:** 2
**Contribution:** 2
**Rating:** 2
**Confidence:** 5

**Summary:**

This work investigates the interplay between model complexity and data complexity in time series forecasting, comparing statistical, deep learning, and foundation models. The study delivers interesting empirical findings—notably, that large foundation models struggle on simple deterministic datasets while excelling on stochastic data. This “bigger isn’t better” insight contributes an important empirical perspective to ongoing debates about the scalability and universality of large models.

**Strengths:**

1. The study provides a broad empirical comparison across a hierarchy of models—from ARIMA and NARMA to deep architectures (e.g., TimesNet, TFT) and foundation models—covering datasets of different complexity levels.

2. The paper raises an important and underexplored question: whether “bigger” models are always better for time series tasks. This theme resonates with the current debate over the universality and efficiency of large-scale foundation models.

**Weaknesses:**

In overall, the theoretical depth and analytical interpretation are limited. The paper largely remains an empirical benchmark rather than offering a principled understanding of why such behavior emerges. Consequently, the contribution falls short of ICLR’s standards for theoretical and conceptual depth.

1. The paper identifies a clear empirical trend—that high-capacity models perform poorly on simple data—but does not provide a theoretical or mechanistic explanation of why this occurs. Although the discussion briefly invokes the approximation-estimation trade-off, it remains purely qualitative, without formal definitions, analytical derivations, or quantitative validation. Consequently, the study is descriptive rather than explanatory, offering limited theoretical insight into the dynamics of model–data interaction.

2. As a benchmark study, the evaluation is narrow in experimental coverage. The results focus mainly on RMSE comparisons across a small set of datasets. Missing are several key analyses that a strong benchmark should include: sensitivity to noise level or data perturbations, robustness under missing data or distribution shift and horizon-length or sampling-frequency sweeps. Without these, the claim of providing a “systematic benchmarking approach” remains overstated.

3. The paper claims to explore how model performance depends jointly on model and data complexity, but this relationship is not formally characterized. There is no scaling analysis or empirical curve demonstrating how error changes as a function of both dimensions. A simple parametric or scaling-law fit could have provided the theoretical clarity currently missing.

4. “Data complexity” is defined conceptually through examples--simple functions vs. stochastic processes--but never quantified. Employing measurable proxies such as permutation entropy, Lempel–Ziv complexity, or spectral flatness would have enabled a more rigorous correlation analysis between model performance and data complexity. The lack of quantification weakens the central thesis.

**Questions:**

Refer to the weaknesses

---

### Note · Authors · 2025-11-16

I have read and agree with the venue's withdrawal policy on behalf of myself and my co-authors.